# Examining How Dog ‘Acquisition’ Affects Physical Activity and Psychosocial Well-Being: Findings from the BuddyStudy Pilot Trial

**DOI:** 10.3390/ani9090666

**Published:** 2019-09-07

**Authors:** Katie Potter, Jessica E. Teng, Brittany Masteller, Caitlin Rajala, Laura B. Balzer

**Affiliations:** 1Department of Kinesiology, University of Massachusetts Amherst, Amherst, MA 01003, USA; 2Last Hope K9 Rescue, Boston, MA 02109, USA; 3Department of Exercise and Sports Studies, Smith College, Northampton, MA 01063, USA; 4Department of Epidemiology and Biostatistics, University of Massachusetts Amherst, MA 01002, USA

**Keywords:** dog ownership, dog walking, physical activity, accelerometry, psychosocial well-being, prospective trial, animal-assisted intervention, dog rescue, foster dog, shelter dog

## Abstract

**Simple Summary:**

Dog owners are more physically active than non-dog owners, but the direction of the relationship between dog ownership and increased physical activity is unknown. In other words, it is unclear whether acquiring a dog causes a person to become more active, or whether more physically active people choose to acquire dogs. Given that regular physical activity is critical for the prevention and management of numerous chronic diseases, research supporting the hypothesis that dogs make people more active could inform programs and policies that encourage responsible dog ownership. In the BuddyStudy, we used dog fostering to mimic dog acquisition, and examined how taking a dog into one’s home affected physical activity and psychosocial well-being. Nearly half of study participants saw large increases in physical activity and nearly three-quarters saw improvements in mood after fostering for six weeks. More than half met someone new in their neighborhood because of their foster dog. Most participants adopted their foster dog after the six-week foster period, and some maintained improvements in physical activity and well-being at 12 weeks. The results of this pilot study are promising and warrant a larger investigation.

**Abstract:**

Dog owners are more physically active than non-dog owners, but evidence of a causal relationship between dog acquisition and increased physical activity is lacking. Such evidence could inform programs and policies that encourage responsible dog ownership. Randomized controlled trials are the ‘gold standard’ for determining causation, but they are prohibited in this area due to ethical concerns. In the BuddyStudy, we tested the feasibility of using dog fostering as a proxy for dog acquisition, which would allow ethical random assignment. In this single-arm trial, 11 participants fostered a rescue dog for six weeks. Physical activity and psychosocial data were collected at baseline, 6, and 12 weeks. At 6 weeks, mean change in steps/day was 1192.1 ± 2457.8. Mean changes on the Center for Epidemiologic Studies Depression Scale and the Perceived Stress Scale were −4.9 ± 8.7 and −0.8 ± 5.5, respectively. More than half of participants (55%) reported meeting someone new in their neighborhood because of their foster dog. Eight participants (73%) adopted their foster dog after the 6-week foster period; some maintained improvements in physical activity and well-being at 12 weeks. Given the demonstrated feasibility and preliminary findings of the BuddyStudy, a randomized trial of immediate versus delayed dog fostering is warranted.

## 1. Introduction

Regular physical activity (PA) reduces the risk of cardiovascular disease, type II diabetes, depression, dementia, and some cancers [1]. For individuals living with one or more chronic conditions, regular PA is key to limiting disease progression, preventing co-morbid conditions, and improving physical function and quality of life [1]. The 2018 Physical Activity Guidelines for Americans recommend adults engage in 150 weekly minutes of moderate-intensity aerobic PA (equivalent in intensity to a brisk walk) to reap these health benefits [2]. The most recent national statistics suggest that fewer than one in two American adults meet this mark [3].

A growing international literature base has examined the relationship between dog ownership and PA levels. As discussed in two recent meta-analyses, several studies have demonstrated that dog owners, on average, are more active than non-dog owners [4], and that dog owners are more likely than non-dog owners to meet PA guidelines [5]. Importantly, multiple studies have examined this relationship in clinical populations, including patients with ischemic heart disease [6] and diabetes [7]. The primary limitation of the literature base is its cross-sectional nature; to date, only two studies have prospectively examined the relationship between dog ownership and PA [8,9]. While it is possible that dog acquisition leads to an increase in PA, it is also possible that more active individuals opt to become dog owners. A recent study reporting that dog owners are more likely to be home owners and have a higher annual household income than non-dog owners [10] may support the latter hypothesis, as socioeconomic status is a consistent correlate of leisure-time PA levels [11]. 

Determining the direction of the dog ownership–human health relationship has important implications. For example, if dog acquisition leads individuals to adopt a more active lifestyle, programming and policies that aim to improve public health might support responsible dog ownership (e.g., encourage pet-friendly lease agreements, provide financial support for veterinary care in low-income communities). Dog ownership could also become a prescriptive tool for physicians to facilitate patient PA, assuming risks are properly considered and mitigated [12]. If renters that can now have a dog in their home or patients that are ‘prescribed’ a dog choose to acquire a rescue dog, then these initiatives could simultaneously improve human health and dog welfare. 

Rigorous prospective studies of the relationship between dog acquisition and PA are needed to inform public health policy and clinical practice. Studies must collect PA data on new owners before and after they acquire dogs, and compare it to data from a group of people that do not acquire dogs. Ideally, a randomized controlled study design would be used to ensure there are no baseline differences between groups that may affect the outcome. Although quasi-experimental (non-randomized) designs and sophisticated analyses [13,14,15,16] can help control for known confounding variables (variables that influence both acquisition of a dog and PA outcomes), there may be unknown confounding variables. To our knowledge, only one study used a randomized design to examine how taking a pet into the home affects pet owner health. In 2001, Allen et al. randomized 48 hypertensive individuals to a pet ownership plus ACE inhibitor condition or an ACE inhibitor only condition [17]. Individuals in the pet ownership group acquired a pet cat or dog at the time drug therapy began. Mental stress tests were conducted in participants’ homes at baseline and 6 months. The researchers concluded that ACE inhibitor therapy alone reduced resting blood pressure, but that social support through pet ownership reduced the psychological response to mental stress. This study did not assess changes in PA.

Given the serious commitment involved in acquiring a pet, random assignment is no longer considered ethical. The purpose of the current, single-arm trial was to test the feasibility of using dog fostering as a proxy for dog acquisition, as dog fostering is a non-permanent commitment that allows for ethical random assignment. If deemed feasible, a randomized trial of immediate versus delayed dog fostering (or some attention control) would allow for rigorous examination of short-term changes in PA that occur upon taking a dog into one’s home. If participants are given the option to permanently adopt their foster dogs after the study period, this approach may also allow for examination of long-term changes in PA. In the BuddyStudy, participants fostered a dog for six weeks and we collected PA, sedentary behavior, and psychosocial outcome data at three time-points to test the feasibility of assessment procedures and examine preliminary effectiveness.

## 2. Materials and Methods 

### 2.1. Sample 

BuddyStudy participants were non-dog owners who were willing and able to be the primary caregiver for a rescue dog for 6–8 weeks. Exclusion criteria included (1) <21 years old, (2) self-reported regular exercise over the past 6 months, (3) lack of reliable source of transportation, (4) presence of any conditions that limit ability to walk, (5) presence of uncontrolled hypertension or diabetes, (6) extensive upcoming travel plans, and (7) dog allergy. Inclusion/exclusion criteria aimed to identify a sample of inactive adults who could safely walk for exercise, and whose living arrangements allowed fostering. All participants were recruited through social media outlets affiliated with the University.

### 2.2. Community Partner 

Dogs involved in this study were fostered through Last Hope K9 Rescue (LHK9), an all-volunteer, all-breed dog rescue organization based in Boston, MA. LHK9 is a foster-based rescue, meaning they rely on foster homes for their dogs and do not have a brick-and-mortar facility. All LHK9 dogs are evaluated and vetted prior to transport from their southern partners in Arkansas, and again prior to entering foster care in New England. Traditionally, LHK9 foster families foster a dog until the dog is adopted, usually between 3–4 weeks, and they are not allowed to adopt their foster dog if there is prior interest from other adopters. Procedures were modified for BuddyStudy participants, who fostered for an extended period of time (minimum six weeks) and were given the option to “foster-to-adopt” as part of the study. 

### 2.3. Study Design

The BuddyStudy was a single-arm feasibility study. After fostering for six weeks, each participant could adopt the dog, transfer the dog to another foster family, or continue fostering until his/her foster dog was adopted into a permanent home. All participants provided written informed consent to participate in the study and signed a contract, which included a liability waiver, to foster with LHK9. This study was approved by the University’s IRB and IACUC.

### 2.4. Procedures

#### 2.4.1. Screening Procedures 

Initial eligibility was determined via an online screening survey (Qualtrics). Preliminarily eligible individuals attended an orientation at the University where study procedures were explained and written informed consent obtained. Individuals were asked to complete an application to foster via the LHK9 website within 48 hours of attending orientation. LHK9 volunteers followed standard screening procedures for prospective foster home applications, including conducting home visits and calling personal references, current or recent veterinarians, and landlords (when applicable). Individuals deemed ineligible to foster were compensated $25. 

#### 2.4.2. Foster Procedures

Approved participants attended a virtual new foster orientation, and then were added to a private online group moderated by LHK9 volunteers. Each week, a foster coordinator would post a list of dogs (including photos, age, breed and any known background information) needing a foster home in New England. Young puppies (less than one year old), dogs with known medical or behavioral issues, or dogs who already had interested adopters waiting were not eligible for the BuddyStudy. After a participant matched with a foster dog, they picked up an 8-week supply of dog food, a slip lead, a six-foot non-retractable leash, dog toys, and a crate at the University. Each LHK9 foster dog was microchipped and fitted with a no-slip martingale collar with identification tag. 

Per Massachusetts state law, all dogs coming into Massachusetts from out-of-state must be isolated in an approved facility for a 48-hour quarantine period before going to a foster home. LHK9 volunteers told participants what day their foster dogs were ready to be picked up at the quarantine facility, typically with 72 hours notice. If a participant was unable to pick up the dog on the pick-up date, they were asked to notify the research team for coverage. Each participant’s 6-week study foster period began the day he/she picked up his/her foster dog.

In addition to the information provided during the virtual orientation, LHK9 provided all BuddyStudy participants with a Foster Information Packet with general dog care information, rescue protocols, and an extensive list of contacts and resources, including a 24/7 emergency line. Participants were asked to post all non-emergency questions in the private online group where a team of volunteers, including LHK9’s medical and training coordinators, could respond and provide support. All preventative care (i.e., flea/tick, heartworm) for the duration of the foster period and any medical issues requiring veterinary care were coordinated and paid for by the rescue. Leash and collar safety were discussed, but no specific instructions regarding dog walking were provided. 

### 2.5. Measures

Data were collected at baseline (pre-foster period), six-weeks (last week of foster period), and 12-weeks (post-foster period). Of note, some participants (adopters) had a dog in their home at 12 weeks and others (non-adopters) did not. 

#### 2.5.1. Feasibility

The primary purpose of the BuddyStudy was to determine the feasibility of using dog fostering as a proxy for dog ownership. Formal feasibility assessments focused on recruitment potential (number of completed applications, proportion of applicants deemed eligible, proportion enrolled), participant attrition (proportion dropping out prior to foster and proportion dropping out after getting the dog), data completeness, significant adverse events (number, type), and percentage of dogs adopted. 

#### 2.5.2. Device-Measured PA and Sedentary Behavior

The ActiGraph wGT3X-BT monitor (ActiGraph, Pensacola, FL, USA) was used to assess PA and sedentary behavior. The ActiGraph is a research-grade triaxial accelerometer deemed valid and reliable in free-living conditions [18,19]. Participants wore the device on an elastic band at their right hip during all waking hours (except when showering/swimming) for Seven consecutive days at all three time points. During each seven-day assessment period, participants logged all leisure-time PA, including dog walking specifically, to provide contextual information about their activity (as accelerometers only provide data on amount of PA, not type). ActiGraph data were processed using Actilife Version 6.13.3. (ActiGraph, Pensacola, FL, USA) to determine steps/day and PA minutes/day spent in each intensity category (sedentary, light, moderate, moderate-to-vigorous [MVPA]) based on the Freedson cut points [20]. A minimum of three weekdays and 1 weekend day with at least 10 hours of wear time was required for inclusion in analyses.

#### 2.5.3. Self-Reported Dog Walking

Three questions from the Dogs And WalkinG Survey (DAWGS) were used to assess self-report dog walking behavior at six weeks among all participants and 12 weeks among adopters [21]. Questions included, “how many days do you walk your foster dog in a typical week?” (0–7 days); “how much time do you spend walking during your typical dog walk? (minutes); “on days you walk your dog, on average how many walks do you go on?” (1–5 or more). If participants reported a range for dog walking duration (e.g., 10–20 min), the lower value was used to calculate the average dog walking minutes/week.

#### 2.5.4. Psychosocial Outcomes 

Stress and depressive symptoms were assessed at all three time points. The 10-item Perceived Stress Scale (PSS) [22], which asks about thoughts and feelings during the last month, was used to evaluate changes in stress. This scale is widely used in behavioral health research and its psychometric properties have been established [23]. Scores can range from 0 to 40 with higher scores indicating more perceived stress. The PSS scale has no standard cut points; as a reference, an average score of 15.5 ± 7.4 was found in a large US sample (n = 968) in 2009 [24]. The 20-item Center for Epidemiologic Studies Depression Scale (CES-D) [25], which asks about feelings and behaviors over the past week, was used to measure changes in depressive symptoms; scores can range from 0 to 60 with scores ≥16 indicating risk for clinical depression. The CES-D has demonstrated high internal consistency, acceptable test–retest stability, concurrent validity, and construct validity when used in general American populations [25]. At 6 weeks, questions used in work by Wood and colleagues [26] were used to assess whether participants got to know people in their neighborhood since starting the study and, if so, whether the foster dog facilitated the interaction and whether the interaction developed into a friendship and/or provided a new source of social support (emotional support, informational support, appraisal support, or instrumental support). Participants also answered open-ended questions about the best part of fostering, the most challenging part of fostering, and the effect of fostering on quality of life.

### 2.6. Statistical Analyses

The purpose of this study was to test feasibility and therefore we did not perform inferential statistics. Quantitative data were summarized using means and standard deviations. Qualitative data from open-ended survey questions were coded by two coders (B.M. and C R.) to identify common themes seen throughout the responses. Each coder independently performed a content analysis to identify themes and categories to organize and refine the data. Following individual analyses, the coders compared their results and resolved any discrepancies prior to making conclusions. Direct quotes (de-identified) were extracted from the data to represent the general themes. Analyses of 12-week outcomes were restricted to participants who adopted their foster dog.

## 3. Results

### 3.1. Feasibility

One hundred and twenty-three individuals applied to participate in the BuddyStudy over the course of 6.5 weeks. About one-quarter of applicants (n = 28; 23%) were deemed preliminarily eligible and invited for study orientation. Eighteen individuals (15% of applicants) completed informed consent to enroll in the study. Twelve received the intervention (i.e., fostered a dog) and eleven completed the study. See Figure 1 for a detailed study flow chart. One participant dropped out of the study within days of picking up her dog, and the dog was transferred to a participant that had not yet matched with a dog. All other participants completed the 6-week foster period and completed 6-week and 12-week assessments. At six weeks, 11 of 11 participants provided valid ActiGraph data; at 12 weeks, 8 of 11 participants provided valid data. There were no significant adverse events. Of 11 rescue dogs in the study, 8 were adopted by their study foster family at the completion of the 6-week study foster period.

### 3.2. Participant Characteristics 

All participants who received the intervention (n = 12) were female (100%) and the majority were non-Hispanic white (83%). The average age was 37.8 ± 16.3 years (range 21–62 years). All had a college degree (67%) or were current college students (33%). The majority of participants reported living in a rural (42%) or suburban setting (50%); of seven participants with a yard, only one had a fenced area for their foster dog. Participants averaged 6932.7 ± 2418.9 steps/day, 33.6 ± 19.7 MVPA minutes/day, and 572.4 ± 65.3 sedentary minutes/day at baseline. The average score on the stress measure (PSS) was 15.0 ± 6.9 and the average score on the depressive symptom measure (CES-D) was 13.9 ± 12.5 at baseline.

### 3.3. PA and Sedentary Behavior

Average steps/day and MVPA minutes/day as measured by the ActiGraph, as well as self-reported dog walking data, are presented by time point in Table 1. At 6 weeks, participants reported dog walking 6.5 ± 0.9 days/week (range 5–7) and increased steps/day by 1192.1 ± 2457.8 and MVPA minutes/day by 12.7 ± 20.9 from baseline. The majority reported 10–15 min walk durations (n = 4) or walks ≥30 min in duration (n = 5). Most participants (n = 10) reported taking more than 1 walk/day on days that they walked their foster dog. Nearly half of the sample (n = 5; 45%) increased their steps by >2000 steps/day and their MVPA by >20 min/day. Individual changes in PA are presented in Figure 2. 

At 12 weeks, participants who still had a dog in their home (n = 8) reported walking 6.8 ± 0.5 days/week (range 6–7). The majority reported that typical dog walks were ≥30 min in duration (n = 3) or that walks were sometimes as short as 5 min (n = 4). Most still reported walking more than once per day (n = 6) on days that they walked their dog. Adopters who provided valid ActiGraph data at this time point (n = 6) maintained an increase of 552.7 ± 3557.1 steps/day and 8.8 ± 18.8 MVPA minutes/day from baseline.

Sedentary behavior data are also presented in Table 1. At 6 weeks, participants decreased sedentary minutes/day by 49.8 ± 41.1. Individual changes in sedentary behavior at 6 weeks are presented in Figure 2. At 12 weeks, participants who still had a dog in the home and provided valid ActiGraph data (n = 6) were sedentary for 56.4 ± 59.6 fewer minutes/day, on average, than baseline. 

### 3.4. Psychosocial Outcomes

Average scores on the stress (PSS) and depressive symptom (CES-D) measures by time point are also presented in Table 1. At 6 weeks, participants reported a reduction of 0.8 ± 5.5 on the PSS and 4.9 ± 8.7 on the CES-D. Additionally, more than half of participants (6/11; 55%) reported meeting someone new in their neighborhood because of their foster dog. While only 1 participant considered someone they met through their foster dog to be a friend, 5 of 11 reported receiving some form of social support through an acquaintance they met through their dog. Individual changes in perceived stress and depressive symptoms are presented in Figure 2. At 12 weeks, participants who still had a dog in their home (n = 8) scored 4.8 ± 7.8 points lower on the stress measure (PSS) and 7.2 ± 12.1 points lower on the depressive symptom measure (CES-D), on average, than at baseline. 

Responses to open-ended survey questions at 6 weeks are summarized in Table 2. When asked to describe the best part of fostering a dog, common themes included feelings of fun, happiness, joy, companionship, and family bonding. The most commonly reported challenge of fostering was the stress and responsibility involved with taking care of a dog. Participants reported that fostering a dog either improved their quality of life or that the impact was mixed; no participants reported that fostering had a predominately negative impact on their quality of life. 

## 4. Discussion

Randomized controlled trials provide the highest level of evidence of treatment effectiveness in clinical research. The purpose of the BuddyStudy was to pilot a novel approach that would allow researchers to investigate the PA and health benefits of ‘getting’ a dog using a randomized controlled design. The BuddyStudy used dog fostering as a proxy for dog acquisition, as fostering is a non-permanent commitment and thereby allows for ethical random assignment. The pilot demonstrated that the design is feasible for implementation on a larger scale, based on the high degree of community interest in the project, high retention among participants who fostered a dog, and absence of significant adverse events. Further, most participants permanently adopted their foster dog, which may allow for the examination of long-term changes in PA with dog acquisition. 

The BuddyStudy pilot also demonstrated the preliminary effectiveness of taking a dog into one’s home for increasing PA. At the end of the six-week foster period, around half of participants increased their steps by >2000 steps/day and MVPA by >20 minutes/day. These are clinically meaningful increases, as previous research has demonstrated that increases in the range of 1000–2000 steps/day have been associated with reduced risk of type II diabetes [27], cardiovascular disease [28,29], and all-cause mortality [30,31,32] and the 2018 Physical Activity Guidelines for Americans [2] recommend adults accumulate 150 MVPA minutes/week to improve health and prevent chronic disease. Based on this BuddyStudy data, we anticipate that a sample size of 136 (68/arm) would be needed to detect, with at least 80% power, a difference of 1192 steps/day (change from baseline) between randomized arms in a full-scale trial. Future studies may wish to match self-reported dog walking data with accelerometer data by timestamp [33] to determine the number of steps and MVPA minutes accumulated specifically during dog walking. It should be noted that some participants decreased their daily steps and MVPA minutes from baseline to six weeks. This may have been due to the onset of cold weather, as baseline assessments were completed in September/October and six-week assessments were completed in December in New England.

Many BuddyStudy participants also decreased sedentary time by >45 min/day. There is growing scientific interest in the health risks of too much sitting [34], and some countries now include sedentary behavior guidelines along with PA guidelines. For example, Australia’s government recommends that adults minimize the amount of time spent in prolonged sitting and break up long periods of sitting as often as possible [35]. Our findings are in line with three recent studies that have examined dog ownership in relation to sedentary behavior. In a large epidemiological investigation, Garcia et al. found that dog ownership was associated with a lower likelihood of being sedentary for ≥8 hours/day among postmenopausal women [36]. In two separate studies with older adults, dog ownership was associated with an average of 21 fewer minutes of sedentary time/day as measured by ActiGraph accelerometers [37] and with fewer sitting events as measured by active PAL monitors [38]. 

The BuddyStudy pilot also demonstrated the preliminary effectiveness of taking a dog into one’s home for improving psychosocial well-being. At the end of the six-week foster period, most participants who fostered a dog reported decreases in depressive symptoms. When asked to reflect on how fostering a dog affected their quality of life, multiple participants mentioned increased joy, fun, and companionship, which may explain improvements in mood. Alternatively, few participants reported reductions in stress at six weeks. This likely reflects the significant responsibility and time commitment involved in fostering a rescue dog, which many participants acknowledged when asked about the most challenging thing about the foster experience. Finally, many participants reported meeting someone new in their neighborhood because of their foster dog. The phenomenon of dogs as social facilitators has been previously demonstrated [26], and may be one of the most important health benefits of dog ownership given the powerful influence of social relationships, or lack thereof, on human health and longevity [39,40].

To date, no randomized trials have examined the influence of dog ‘acquisition’ on dog owner PA. Two non-randomized trials have collected self-report PA data on individuals before and after they acquired a dog, and compared it to data from individuals who did not acquire a dog; both found increases in self-reported recreational walking among new dog owners [8,9]. A large randomized trial of immediate versus delayed fostering that employs device-based PA assessment would build on these studies in two key ways: by better controlling for confounding variables that might explain observed increases in PA (with randomization) and by improving the precision of PA assessment (with accelerometry and a larger sample). A larger sample would also allow for the examination of previously demonstrated correlates of dog walking (e.g., owner income, sense of obligation to walk the dog [41]) as moderators and mediators of change in PA.

A number of lessons were learned from the BuddyStudy pilot and should be considered prior to scale-up. First, given the extensive foster screening process and significant commitment that fostering requires, researchers should anticipate substantial attrition between the informed consent and dog matching steps. Second, running the trial in small waves (n = 5–8) may help to not overwhelm the rescue organization and also account for seasonality, which is likely to impact dog-related PA. Third, future trials should be scheduled to avoid major holidays. In the BuddyStudy pilot, we had to board a foster dog for a week after Thanksgiving coverage plans fell through last minute. Finally, and more generally, the pilot taught us that, while feasible, this study is logistically challenging and will require a sizable research team and budget to be properly conducted. In addition to traditional costs of conducting a randomized trial, the budget should cover costs relating to dog transportation, quarantine boarding, preventive medical care for all study dogs, dog foster supplies (e.g., crates, food, leashes), and reimbursement for short-term boarding, training, and medical care, like dewormer medication and vet visits (all of which are typically covered by rescue organizations). 

Finally, while the BuddyStudy study design ultimately uses dog fostering to examine how ‘acquiring’ a dog influences PA and health, it simultaneously raises awareness about the need for foster homes to reduce shelter euthanasia rates and recruits new volunteers for the cause. An estimated 6–8 million cats and dogs enter US animal shelters each year and 3 million are euthanized [42]. Across the US, shelters are filled with dogs that need to be cared for and exercised by volunteers. Foster families are also needed to make room in the shelters for more surrendered and abandoned dogs. Foster-based dog rescue organizations, like LHK9 Rescue, transport dogs from shelters in overpopulated parts of the country to areas that tend to have fewer dogs for people to adopt locally. The more foster families these organizations have, the more dogs they can pull in to rescue. All 11 rescue dogs involved in the BuddyStudy pilot were permanently adopted, eight by their study foster parents.

The primary limitations of this study are the small sample size and lack of control group, which limit our ability to draw conclusions about effectiveness. The study sample was also highly homogenous (100% women; 83% non-Hispanic white). Future studies, including multi-site trials, could examine effectiveness in a more diverse population and across different geographic regions. It should be noted that the vast majority of animal welfare volunteers are women, and therefore this approach may ultimately appeal more to women [43]. Importantly, women are less active and therefore in greater need of PA intervention than men [3]. Finally, our two primary psychosocial variables were assessed over different time frames (past week for depressive symptoms, past month for perceived stress) and future studies should consider using measures with more similar time frames. Strengths of this study include the innovative approach taken to address a critical gap in the literature, which required a novel University-Dog Rescue partnership. The use of accelerometry to measure PA is also a strength, as the literature on dogs and PA is heavily reliant on self-report PA measures prone to recall and social desirability biases. 

## 5. Conclusions

Given the demonstrated feasibility and preliminary findings of the BuddyStudy pilot, a randomized trial of immediate versus delayed dog fostering is warranted and will provide the most rigorous evidence to date of the effects of dog acquisition on human PA and health. Given the major disease and economic burden caused by physical inactivity [44], as well as the ongoing pet homelessness epidemic [42], this line of research has significant societal implications. Demonstration of a positive causal relationship between dog acquisition and increased PA could lead to the testing and implementation of evidence-based public health policies and programs that encourage responsible dog ownership.

## Figures and Tables

**Figure 1 animals-09-00666-f001:**
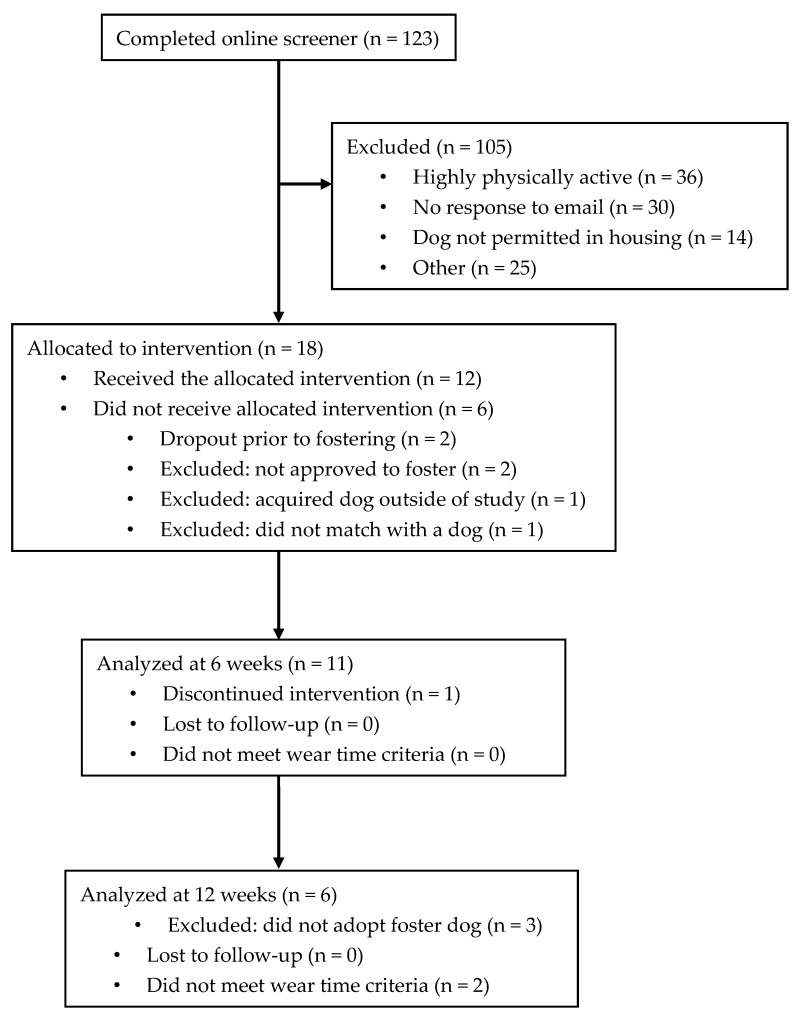
BuddyStudy flow chart.

**Figure 2 animals-09-00666-f002:**
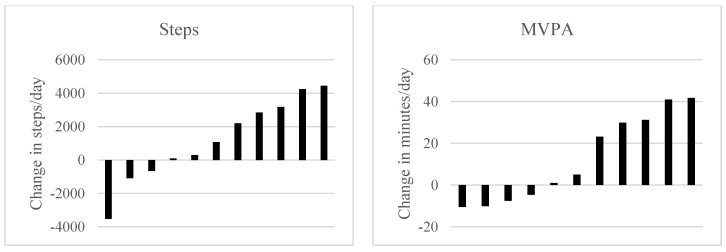
Individual participant changes from baseline to 6 weeks in physical activity, sedentary behavior, and psychosocial outcomes in the BuddyStudy (n = 11). Black bars indicate positive change is better; red bars indicate negative change is better. Physical activity and sedentary behavior were measured via ActiGraph accelerometer. Abbreviations: MVPA = moderate-to-vigorous physical activity; PSS = Perceived Stress Scale; CES-D = Center for Epidemiologic Studies- Depression scale.

**Table 1 animals-09-00666-t001:** BuddyStudy physical activity (PA) and psychosocial outcomes by time point.

Outcome	Baseline (*n* = 11) ^1^	Week 6 (*n* = 11)	Week 12 (*n* = 8)^2^
Device-measured PA			
Daily steps	6976.6 ± 2532.0	8168.7 ± 3827.8	7528.1 ± 4356.9^3^
MVPA minutes per day	35.3 ± 19.7	48.0 ± 33.6	37.8 ± 32.2^3^
Sedentary minutes per day	574.0 ± 65.3	524.3 ± 68.3	512.3 ± 94.3^3^
Self-reported dog walking			
Days w/ at least 1 walk	-	6.5 ± 0.9	6.8 ± 0.5
Minutes per typical walk	-	23.0 ± 16.8	18.1 ± 17.3
Minutes of dog walking per week	-	340.3 ± 243.7	242.5 ± 196.1
Depressive symptoms (CES-D; scale range 0–60)	14.4 ± 13.0	9.5 ± 10.0	7.9 ± 8.9
Perceived stress (PSS; score range 0–40)	15.6 ± 6.9	14.7 ± 5.8	10.0 ± 7.4
Social facilitation			
*n* (%) met person in neighborhood through foster dog	-	6 (55%)	-
*n* (%) consider person met through foster dog to be a friend	-	1 (9%)	-
*n* (%) received social support from person met through foster dog	-	5 (45%)	-

Results reported as mean ± standard deviation unless otherwise noted. ^1^ Only includes baseline data from participants who completed the study. ^2^ Adopters, only. ^3^ n = 6 (n = 2 adopters had invalid ActiGraph data at 12 weeks). Abbreviations: MVPA = moderate-to-vigorous physical activity; CES-D = Center for Epidemiologic Studies-Depression scale; PSS = Perceived Stress Scale.

**Table 2 animals-09-00666-t002:** Open-ended participant responses at 6 weeks in the BuddyStudy.

Question/Prompt	N Endorsing ^1^	Exemplary Quotes
“What has been the best thing about fostering a dog?”		
Fun/happiness/joy	5	*“My pup is a continuous source of happiness in my life - coming home is much more enjoyable than it ever has been. To know she is always at home waiting for me gives me a sense of joy and comfort.”*
Companionship	3	*“The continuous love and kisses and snuggles! He is a constant companion that always cheers me up.”*
Family bonding	3	*“The best thing about fostering X has been our family working together to care for him.”*
“What has been the most challenging thing about fostering a dog?”		
Stress/responsibility	4	*“In a busy household, I already feel like I have a pile of people to care for, be available for, keep healthy, feed, etc. Adding another one can sometimes feel overwhelming (especially when she wasn’t feeling well)…”* *“The learning curve of understanding my foster dogs needs and how to juggle that with my own needs has been tricky at times.”*
“Please write a brief reflection on how fostering a dog has affected your quality of life.”		
Improved quality of life	5	*“It’s improved my quality of life. He’s brought a lot of joy into my life, and I think I’m less stressed since he’s around.”* *“Fostering has provided me with a happier and healthier lifestyle. Coming home to a pup that is excited to see you is the best feeling. Fostering has also encouraged me to go on many more walks and seek a more active lifestyle.”*
Mixed impact on quality of life	3	*“It has been overwhelming, exciting, stressful, relaxing, wonderful and difficult.”* *“It has been very mixed. I have been happy to have his company, and gratified to help him learn, but it has been very difficult to do anything but take care of him for the last six weeks…”*

^1^ Only themes endorsed by at least three participants are included in the table.

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
