# Peer review of "Examining How Dog ‘Acquisition’ Affects Physical Activity and Psychosocial Well-Being: Findings from the BuddyStudy Pilot Trial"

_animals, 2019, doi:10.3390/ani9090666_

Round 1
Reviewer 1 Report
Thank you for the opportunity to review this manuscript.
This manuscript addresses an interesting and important topic. It is notable that efforts have been made in identifying a way allowing for RCT studies to be conducted. From my perspective the major limitation of this study relates to its limited sample size and the lack of inferential statistics. Notwithstanding, I believe it is worth publishing as it will certainly stimulate further research.
My detailed comments regarding each section of the manuscript can be found below. I congratulate the authors on their efforts and wish them luck on their work.
Introduction: It is clear and straight to the point. Also it provides relevant background. However, I consider some important references are missing and should be added to the manuscript. Please see below.
Line 52-53: "Numerous studies have demonstrated that dog owners, on average, are more active 52 than non-dog owners [4], and that dog owners are more likely than non-dog owners to meet PA 53 guidelines [5]."
What about studies showing no such association? Are there any? I think they should also be highlighted here.
Also, some of the conducted studies addressing the association between dog ownership and physical activity have focused on clinical populations. I believe this is of major importance to be highlighted in this paper. Some examples:
Dunn, S. L., Sit, M., DeVon, H. A., Makidon, D., & Tintle, N. L. (2018). Dog Ownership and Dog Walking: The Relationship With Exercise, Depression, and Hopelessness in Patients With Ischemic Heart Disease. Journal of Cardiovascular Nursing, 33(2), E7-E14.
Riske, J., Janert, M., Kahle-Stephan, M., & Nauck, M. A. (2019). Owning a Dog as a Determinant of Physical Activity and Metabolic Control in Patients With Type 1 and Type 2 Diabetes Mellitus. Experimental and Clinical Endocrinology & Diabetes.
Line 63: "remove breed-specific legislation" - what do the authors mean by that? Please clarify.
Line 64: "Dog ownership could also become 64 a prescriptive tool for physicians to facilitate patient PA." In this respect it might be interesting to comment on Kate Hodgson's work (one example below).
Hodgson, K., Barton, L., Darling, M., Antao, V., Kim, F. A., & Monavvari, A. (2015). Pets' impact on your patients' health: leveraging benefits and mitigating risk. The Journal of the American Board of Family Medicine, 28(4), 526-534.
Line 67: "Alternatively, if individuals who are already active choose to become dog owners, then policies and programming that support dog ownership are not likely to increase PA levels further." Why not? What are the authors' arguments for this claim? Any reference(s) to support it?
Line 74: "Although quasi-experimental (non-74 randomized) designs and sophisticated analyses can help control for known confounding variables". Please add some references regarding these sophisticated analyses.
Line 78: "In 1991, Allen et al. randomized 48 hypertensive 78 individuals to a pet ownership plus ACE inhibitor condition or an ACE inhibitor only condition [10]." It was in 2001. Please correct the date.
Methods: This section is adequately described. Yet, some additional details/clarifications are needed.
Line 170: "Participants wore the device on an elastic band at their right 170 hip during all waking hours (except when showering/swimming) for 7 days at all three time points." Please clarify how these 7 days were selected.
Line 172: "During each 7-day assessment period, participants logged all leisure-time PA, including dog walking 172 specifically, to provide context." Not clear. Please rephrase or further explain.
Regarding the PSS: it seems (from the literature) that the questions in this scale ask about the respondents' feelings and thoughts during the last month. This information should be added in the manuscript. Also, according to the literature, scores ranging from 0-13 would be considered low stress, scores ranging from 14-26 would be considered moderate stress; scores ranging from 27-40 would be considered high perceived stress. This might also be worth adding in the manuscript and results could highlight observed changes in categories.
Regarding the Depression scale: it seems that the questions in this scale ask about the respondents' feelings during the last week. This information should be provided to the reader. Also, I think the authors should refer to this difference between the scales (last month vs last week). Why having selected these two scales given such a difference?
Results and discussion sections: results are clearly presented. Notwithstanding, I think it would be a valuable add to the manuscript if the authors could also present a calculation, based on their results, of the sample size needed in an future RCT. In particular, I am concerned about the fact that "only about one-quarter (of 123 individuals) were deemed preliminarily eligible and invited for study orientation". How long did it take to recruit 123 participants? Since this is a feasibility study, I think readers may look for more information about the sampling efforts required in an RCT - which is the obvious extension of this study. Also, it would be interesting to further discuss the fact that all participants who received the intervention (n=12) were female (100%) and that the majority of participants reported living in 223 a rural (42%) or suburban setting (50%). It is recognized that there may be increased needs for physical activity among individuals living in urban contexts. How to have a more balanced sample population in an RCT study? Given the nature of this study - a feasibility study - I would suggest authors to further discuss these issues.
Reviewer 2 Report
This paper is well-written and has the potential to significantly contribute to the literature. The intervention approach is novel and has the potential for high impact.
Methods:
-Please add reliability/validity data for your measures.
-What activity cut points were used to determine intensity of PA?
-More information about the determination of qualitative themes is needed- what strategy was used? Were there two coders?
Discussion:
-Inclusion of this paper in the discussion would address your limitation of not matching up dog walking time via the accelerometer timestamps: Richards, E., Troped, P., & Lim, E. (2014). Assessing the intensity of dog walking and impact on overall physical activity: A pilot study using accelerometry. Open Journal of Preventive Medicine, 4(7), 523-528.
-Discussion of the living environment of the participants (i.e. presence of a yard, fence, sidewalks) would be helpful to understand how that might impact the walking behaviors of the foster participants.
-It is important to also hypothesize or discuss why some participants increased their sedentary time or decreased their PA at 6 weeks? Were these the same participants who had increases in stress or depressive symptoms?
